# Wheat Metabolite Interferences on Fluorescent *Pseudomonas* Physiology Modify Wheat Metabolome through an Ecological Feedback

**DOI:** 10.3390/metabo12030236

**Published:** 2022-03-09

**Authors:** Laura Rieusset, Marjolaine Rey, Florence Wisniewski-Dyé, Claire Prigent-Combaret, Gilles Comte

**Affiliations:** Ecologie Microbienne, Université Claude Bernard Lyon1, Université de Lyon, CNRS UMR-5557, INRAe UMR-1418, VetAgroSup, 43 Boulevard du 11 Novembre 1918, 69622 Villeurbanne, France; marjolaine.rey@univ-lyon1.fr (M.R.); florence.wisniewski@univ-lyon1.fr (F.W.-D.); claire.prigent-combaret@univ-lyon1.fr (C.P.-C.); gilles.comte@univ-lyon1.fr (G.C.)

**Keywords:** *Pseudomonas*, wheat, secondary metabolites, plant–bacteria interaction, metabolomics, ecological feedback

## Abstract

Plant roots exude a wide variety of secondary metabolites able to attract and/or control a large diversity of microbial species. In return, among the root microbiota, some bacteria can promote plant development. Among these, *Pseudomonas* are known to produce a wide diversity of secondary metabolites that could have biological activity on the host plant and other soil microorganisms. We previously showed that wheat can interfere with *Pseudomonas* secondary metabolism production through its root metabolites. Interestingly, production of *Pseudomonas* bioactive metabolites, such as phloroglucinol, phenazines, pyrrolnitrin, or acyl homoserine lactones, are modified in the presence of wheat root extracts. A new cross metabolomic approach was then performed to evaluate if wheat metabolic interferences on *Pseudomonas* secondary metabolites production have consequences on wheat metabolome itself. Two different *Pseudomonas* strains were conditioned by wheat root extracts from two genotypes, leading to modification of bacterial secondary metabolites production. Bacterial cells were then inoculated on each wheat genotypes. Then, wheat root metabolomes were analyzed by untargeted metabolomic, and metabolites from the Adular genotype were characterized by molecular network. This allows us to evaluate if wheat differently recognizes the bacterial cells that have already been into contact with plants and highlights bioactive metabolites involved in wheat—*Pseudomonas* interaction.

## 1. Introduction

Wheat roots exude a large diversity of molecules, including both primary (i.e., sugars, amino acids, organic acids, etc.) and secondary metabolites (i.e., phenolic derivatives, alkaloids, terpenoids, etc.) [1]. The presence of these molecules in the soil zone, in contact with the roots (i.e., the rhizosphere), participates in the selection of soil microbial taxa and constitution of the rhizomicrobiota [2]. Some members of the rhizomicrobiota possess key genetic functions for plant health and growth [3]. Indeed, some are able to promote plant growth and health through the co-expression of different plant-beneficial functions [4,5]. These microbial functions contribute to the promotion of plant nutrition (i.e., through phosphate solubilisation, siderophore production, nitrogen fixation, etc.) and protection (i.e., through production of plant hormones, bioactive secondary metabolites, volatiles organic compounds, or enzymes modulating the plant hormonal pathways) [6]. Plant protection against pathogen attacks relies on three different strategies: induction of the plant systemic defense (ISR), competition against possible pathogens, and antagonism via the production of antimicrobial compounds [7].

Plant-beneficial functions often involve secondary metabolites [7,8]. Some strains producing bacterial secondary metabolites (BSM) are involved in plant protection against pathogens through direct antimicrobial activity or promotion of plant defense [9]. Plant-beneficial bacteria are also able to produce bioactive metabolites that exhibit a signaling activity on plant biosynthesis pathways and influence plant growth [6]. Yet, one of our previous studies [10] showed that wheat root extracts containing bioactive metabolites can significantly influence the production of secondary metabolites by several rhizospheric *Pseudomonas* strains. We showed that root extracts of the wheat genotypes Adular and Bordeaux (at a concentration of 50 µg·mL^−1^) could alter the accumulation of 39% and 29% of the secondary metabolites produced by *Pseudomonas ogarae* F113 and 38% and 28% of those produced by *Pseudomonas chlororaphis* JV395B, respectively. More specifically, this study revealed a differential accumulation of secondary metabolites involved in the plant-beneficial activity of the JV395B strain, such as *N*-acyl homoserine lactones (AHLs) or phenazines. AHLs are mediators that can have an inter-kingdom signaling effect. Besides their involvement in bacterial communication networks, they could be recognized by plant receptors and lead to the modification of plant gene expression [11,12] and positive effects on plant growth [13,14]. The increase in AHL production in the conditioned cells can be correlated with an increase in phenazine production [10]. Indeed, the conditioned JV395B cells produced a higher amount of phenazines, especially hydroxyphenazine (OH-PHZ) and dihydroxy-phenazine-1-carboxylic acid (di-OH-PCA) derivatives [10]. Phenazines are known to be major mediators of plant–bacteria interaction within the *P. chlororaphis* subgroup [15]. They play a major role in the survival of *P. chlororaphis* in the rhizosphere of wheat and protection of plant from pathogens, both via antagonistic activities, such as PCA on *Botrytis cinerea*, and induction of induced systemic resistance (ISR) [15,16,17]. In strain F113, the presence of the extracts did not lead to an increased content of antimicrobial compounds. On the contrary, it led, for instance, to a decrease in the production of DAPG [10], which is a major compound responsible for the biocontrol activity of F113 [6]. Nevertheless, at sub-inhibitory concentrations, this molecule is known to have a signaling effect on plant roots by modulating certain biosynthetic pathways, such as the auxin hormonal pathway [18,19]. It is, therefore, conceivable that, at low concentrations, this molecule could induce metabolic changes in Adular roots. Moreover, the conditioning of strain F113 cells significantly increased the production of compounds not yet studied for their potential plant-beneficial activity, such as pyridine-2,6-thiocarboxylic acid (PDTC) or lipophilic nitrogen derivatives [10].

The production of these bacterial secondary metabolites in the rhizosphere may subsequently lead to specific physiological responses in the host plant [19,20]. Not all plant genotypes interact in the same way with *Pseudomonas* plant growth-promoting rhizobacteria (PGPR) [21]. Recently, we showed that wheat colonization by F113 and expression of 2,4-diacetylphloroglucinol biosynthetic genes (*phl*) was overall higher on ancient genotypes than on modern genotypes [22]. Moreover, F113 best improved the growth of wheat genotypes that showed the highest level of *phl* expression [22].

Thus, one can assume that *Pseudomonas* cells that have previously been in contact with wheat root extracts will produce different secondary metabolites in the rhizosphere and, consequently, will then differentially impact the plant physiology, compared to cells that have not been previously in contact with the plant. To test this hypothesis, we specifically preconditioned the two strains *P. ogarae* F113 and *P. chlororaphis* JV395B with root extracts of the Adular and Bordeaux wheat genotypes, resulting in three different conditionings per strain (i.e., unconditioned, Adular-conditioned, and Bordeaux-conditioned) (Figure 1a). Each of the six different types of conditioned bacterial cultures was then inoculated onto each of the Adular and Bordeaux wheat genotypes, resulting, along with the uninoculated condition, in a total of 14 different conditions (Figure 1b). We then monitored the growth of the wheat genotypes and performed a root metabolite profiling, in order to evaluate the impact of inoculating conditioned cells. Our novel cross-metabolomics approach allows us to assess the ecological feedback between wheat genotypes and plant-beneficial *Pseudomonas*, as well as to identify the plant and bacterial metabolites involved in plant–bacteria interactions.

## 2. Results

### 2.1. Conditioning of Pseudomonas Cells with Wheat Extracts and Evaluation of Inoculation with Conditioned and Unconditioned Bacterial Cells on Wheat Genotype Growth

After being grown in the presence (i.e., conditioned) or absence (i.e., unconditioned) of wheat root extracts, the bacterial cells were inoculated on seeds of the Adular and Bordeaux genotypes (Figure 1).

Controlling the size of the inoculum allowed us to confirm that the same population size (2 × 10^7^ CFU·ml^−1^) was inoculated in each condition (Appendix A). In parallel, an aliquot of the inoculum was subjected to profiling of secondary metabolites (Figure 1a). Adular and Bordeaux plant extracts induce the modification of the production of more than half of F113 and JV395B metabolites (i.e., 53.5% and 54.2%, respectively). Furthermore, as reported in our previous work [10], the production of compounds involved in *Pseudomonas* plant-beneficial properties are altered in conditioned bacterial cells. For strain F113, the production of DAPG decreases, while the production its precursor the mono-acetylphloroglucinol (MAPG) increase. Moreover, PDTC and other lipophilic nitrogen derivatives (i.e., *m*/*z*: 238, 280, and 264) strongly increases in presence of both Adular and Bordeaux genotype extracts (Appendix A). It can also be noted that the Adular genotype has a more marked effect than the Bordeaux genotype. With respect to JV395B, the production of AHLs, phenazine, and pyrrolnitrin compounds were altered in same proportion in Adular and Bordeaux conditioned bacterial cells. The production of all AHLs derivatives is increased, that of pyrrolnitrin is decreased, and not all phenazine derivatives are impacted in the same way. (i.e., production of hydroxy-phenazine and dihydroxyphenazine-1-carboxylic acid increased, while the production of phenazine-1-carboxylic acid decreased) (Appendix A). These results confirm a differential production of secondary metabolites between conditioned and unconditioned strains.

### 2.2. Impact of the Inoculation of Conditioned and Unconditioned Bacterial Cells on Plant Growth

The conditioned and unconditioned cells were inoculated onto seeds of both wheat genotypes, which were then grown for 21 days in non-sterile soil. (Figure 1b). A measurement of the size of the aerial parts was performed throughout the growth of the plants (Appendix A), as well as measurements of the dry biomasses of the aerial and root parts, at the end of the experiment (Appendix A). These measurements did not reveal any significant effect of inoculation on the growth of the two wheat genotypes (Appendix A). Nevertheless, a slight increase in root biomass of the Bordeaux genotype, in the presence of the JV395B Bordeaux-conditioned cells, can be highlighted (Appendix A).

### 2.3. Impact of the Inoculation of Unconditioned Bacterial Cells on Root Metabolism of Two Wheat Genotypes

A first principal component analysis, gathering data from all 14 conditions, was constructed (Appendix A) and showed that the main difference between conditions is the difference in metabolism between the Adular and Bordeaux genotypes. We, therefore, chose to treat the two wheat genotypes separately. We evaluated the impact of each of the two unconditioned cells on the secondary metabolism of Adular and Bordeaux. For this, we performed a multivariate statistical analysis, by PLSDA, for each genotype uninoculated and inoculated with each of the two strains (Figure 2). The PLSDA models were validated by a permutation test [23] and had Q_2_ parameters of 0.80 and 0.69 for the Adular and Bordeaux genotypes, respectively, which showed the significance of the models. The models identified discriminant plant metabolites involved in the separation between conditions. A univariate analysis was then performed on these metabolites (Kruskal-Wallis test; *p* value < 0.05; *N* = 7), allowing us to highlight the metabolites whose amount in roots was significantly modified by inoculation. This approach showed that among the 1060 metabolites detected in wheat extracts, 208 (19.6%) and 149 (14.1%) metabolites from Adular and Bordeaux, respectively, were differentially produced when the plants were inoculated. PCAs performed on the discriminating compounds of each genotype showed that all conditions were separated on the two main axes, which explains a total of 47.6 and 44.4% of the variability in the data in Adular and Bordeaux, respectively (Figure 2).

The Adular and Bordeaux genotypes appeared to respond differently to strain inoculation. In the Adular genotype, axis 1 (28.0% of the variability) separates the root metabolism of plants inoculated with strain JV395B from those inoculated with strain F113, showing a potential specificity of response of Adular (Figure 2a). Indeed, among the vast majority of discriminating metabolites, compared to the uninoculated condition (NI), inoculation with each of the strains did not impact the same metabolites (i.e., only 37 common metabolites) (Figure 2c). Furthermore, 28 metabolites displayed a differential accumulation, following both inoculations, but not with the NI condition (Figure 2c). The distribution of the discriminating metabolites also showed that on the Adular genotype, the inoculation of JV395B, rather, led to a decrease in the accumulation of discriminant metabolites, whereas F113 rather led to an increase, compared to uninoculated plants. The Bordeaux genotype seems to be less affected by the *Pseudomonas* bacterial inoculations. It showed 50 fewer discriminating compounds than the Adular genotype (i.e., with 149 discriminant metabolites for Bordeaux versus 208 for Adular). For the Bordeaux genotype, axis 1 (30.9% of variability) separates the bacterial inoculations from the NI condition, and axis 2 (13.5% of variability) separates the bacterial treatments between each other (Figure 2b). The distribution of metabolites showed that inoculation of strain JV395B triggered more changes (98 metabolites) than strain F113 (80 metabolites), compared to the NI condition (Figure 2d). The analysis of the data shows an important specificity of interaction between strains and genotypes. For this reason, the impact of conditioned and unconditioned strains on root metabolism was evaluated separately for each genotype x strain interaction.

### 2.4. Impact of Inoculation of Conditioned Bacterial Cells on Root Metabolism of Two Wheat Genotypes

PLSDAs performed on the dataset showed significant models for the Adular × JV395B, Adular × F113, and Bordeaux × JV395B interactions with Q_2_ parameters of 0.82, 0.41, and 0.62, respectively. No model allowing a meaningful separation of conditions could be generated for the Bordeaux × F113 interaction.

For interactions with a significant PLSDA pattern, PCAs were performed on significantly discriminating compounds to visualize differences between conditions (Figure 3a, Figure 4a and Figure 5a). Heatmaps, associated with a clustering of the closest conditions, allowed us to visualize the distribution of the discriminating metabolites according to their accumulation in the different conditions. They were associated with each PCA, in order to visualize the response of plant metabolites to bacterial inoculation by conditioned or unconditioned cells. Metabolites showing the same changes in their amounts, under the compared conditions, were distributed within specific clusters (Figure 3b, Figure 4b and Figure 5b).

The Adular × JV395B (AJ) interaction is the one for which PLSDA showed the highest Q_2_ parameter value and the most robust separation between conditions (Figure 3). PCA highlighted that, first, the separation of all conditions is mostly along a gradient along axis 1, which accounts for 29.2% of the variability. Second, the NI condition is the best separated condition from the others. Finally, plants inoculated with the conditioned JV395B cells (AJA and AJB) have metabolisms more distant from the NI condition than those inoculated with the unconditioned JV395B cells (AJC). The distribution of metabolites (Figure 3b) showed that the majority of metabolites are distributed in the first ten clusters (i.e., clusters 1A to 5A and 1B to 5B), representing the plant metabolites for which the main differences exist, with respect to the NI condition. Among these compounds, the majority showed a reduced accumulation when the plant is inoculated with JV395B. The decrease of accumulation, however, can occur in different ways, depending on the conditions. Indeed, only the metabolites grouped in cluster 1A display a reduced accumulation by the conditioned and unconditioned JV395B cells. The accumulation of the metabolites classified in the other clusters (2A, 3A, 4A, and 5A), representing 112 (40.4%) of the 277 discriminating metabolites, decreased in significant different proportions, depending on whether the strain is conditioned or not. Cluster 2A represents metabolites significantly modified only in the presence of conditioned cells whereas clusters 3A and 4A group metabolites, whose accumulation is affected only in the presence of strains conditioned, respectively, by one of the Adular or Bordeaux genotype, respectively; finally, cluster 5A displays metabolites whose accumulation is modified only in the presence of the unconditioned JV395B cells. The same cluster system was used for metabolites whose production is increased by JV395B inoculations (i.e., 2B, 3B, 4B, and 5B). For metabolites whose accumulation is increased or decreased, compared to the NI condition, the clusters with the highest number of metabolites are clusters 2A and 2B, respectively, showing that the changes in plant metabolites are more important when the JV395B cells are conditioned. Finally, the last two clusters (i.e., 6 and 7) show the metabolites for which no main difference is highlighted, when treatments are compared to the NI condition. Cluster n°6 contains the metabolites for which the main difference is between inoculation of the conditioned and unconditioned cells of JV395B. Cluster 7 represents the metabolites for which the main difference is between inoculations on Adular vs. Bordeaux conditioned JV395B cells (AJA and AJB). This cluster represents 44 (15.8%) of 277 the discriminating metabolites and shows that the production of some compounds can be significantly modulated, according to the wheat genotype used for cell conditioning.

For the Adular × F113 interaction, PCA showed that the NI condition is the most distant from the others (Figure 4a). We also find a gradient along axis 1 that explains 29.5% of the variability, with a greater impact of Adular conditioned F113 cells on root metabolism than other conditions. The inoculation of F113 cells conditioned with Bordeaux extracts (AFB) is also separated from the other conditions along axis 2, which explains 22% of the variability. It can also be noted that this condition shows the greatest variability between replicates (Figure 4a). The discriminant metabolites were grouped on a heatmap, according to the same cluster system as before (Figure 4b). Similarly, to the Adular × JV395B interaction, the majority of metabolites is ranked in the top ten clusters and, thus, displays a differential accumulation between inoculated and non-inoculated conditions. In contrast to the inoculation with strain JV395B, the accumulation of metabolites is mostly increased by inoculation with strain F113. In this F113 × Adular interaction, the condition that causes the strongest metabolic change is the AFB condition (i.e., Bordeaux-conditioned F113 cells) represented by clusters 4B and cluster 8 (i.e., cluster 8 showing the metabolites whose behaviour is quite similar between the NI and AFB conditions, but distinct from other conditions). These two clusters underline the particular impact of the Bordeaux-conditioned F113 cells on the Adular genotype.

Finally, the statistical analysis performed on the Bordeaux × JV395B interaction showed that the Bordeaux genotype has 130 fewer discriminating metabolites than the Adular × JV395B interaction; thus, this genotype seems to be less impacted by JV395B inoculation (Figure 5). Nevertheless, PCA showed that, as for the other interactions, the NI condition is the most distant condition from the others, with a separation along axis 1 that explains 22.8% of the variability (Figure 5a). The condition representing the metabolism of the Bordeaux genotype inoculated with Adular-conditioned JV395B cells (BJA) is also separated from the others along PCA axis 2, which explains 14.5% of the variability. Contrary to what is observed for the Adular genotype, there is no gradient of responses between the JV395B cells conditioned by the two genotypes and the unconditioned cells. When looking at the heatmap associated with the PCA (Figure 5b), the BJA condition is the most separate from the other conditions, confirming the atypical behaviour of the Bordeaux genotype inoculated by Adular-conditioned JV395B cells. Indeed, 32 (21.7%) of the 147 discriminating metabolites are classified in clusters 3A and 3B, showing that their accumulation is specifically increased or decreased in the BJA condition, compared to the NI condition.

### 2.5. Impact of Inoculation of Conditioned Bacterial Cells on the Production of Bioactive Secondary Metabolites in the Adular Genotype

Results of untargeted metabolomic analysis of inoculated and uninoculated wheat roots showed that the Adular genotype responds more markedly to inoculation of *Pseudomonas* strains, particularly when cells are previously conditioned with wheat root extracts. We, therefore, sought to go further to characterize the response of this genotype to *Pseudomonas* inoculation. For this purpose, a molecular network approach coupled with a statistical analysis was done (Kruscal-Wallis univariate non-parametric test, FDR correction, *p* value < 0.05), in order to highlight the discriminating compounds on the network. This method allowed us to detect the discriminant metabolites belonging to known chemical families (Figure 6). The molecular network was performed on 311 ions with good quality fragmentation. Of these ions, 72 could be grouped in families of more than two compounds. The molecular network allows the identification of three main families of molecules: hydroxycinnamic acids, benzoxazinoids, and flavonoids. Phenylalanine is also represented on the network. Finally, four others, as yet unknown groups of compounds with common signatures, have also been identified (Figure 6). Discriminant compounds could be highlighted in six out of the seven groups, particularly in those representing known metabolite families. Nodes represent the discriminant metabolites after *P. ogarae* F113 (purple) and *P. chlororaphis* JV395B (yellow) inoculation. The heatmap, associated with every discriminant compound, makes it possible to visualize under which conditions the metabolites are differentially produced and to which cluster they belong on the untargeted analysis (Figure 3a, Figure 4a and Figure 5a).

Nine compounds of the benzoxazinoid family could be highlighted on the array (nodes in Figure 6). The majority of the benzoxazinoid derivatives are modified in their accumulation following inoculation. Four of them were specifically modified by the JV395B inoculation, one by the F113 inoculation, and two by both inoculations. The inoculation of JV395B induced a significant decrease of BZX-Glc, depending on whether the cells were conditioned or not. Only the accumulation of DIBOA-Glc was significantly decreased after inoculation of the conditioned and unconditioned cells. The amount of DHBOA-Glc and DIMBOA-Glc was significantly decreased in the AJA and AJB conditions, with a more pronounced decrease for DIMBOA-Glc in the AJA condition. Finally, HMBOA-Glc was significantly decreased only in the AJB condition. Inoculation of JV395B also led to a modification of DIMBOA and HDMBOA accumulation. For these BZX, the impact of inoculation of the conditioned and unconditioned cells was more contrasted. The DIMBOA content was decreased under AJA and AJC conditions, while the HDMBOA content was inversely impacted, depending on whether the JV395B cells were conditioned or not. F113 inoculation had little impact on BZX-Glc. On the other hand, it led to a significant increase in non-glycosylated benzoxazinoids (BZX) and MBOA content. The inoculation of F113 also showed differences, whether the cells were conditioned or not. Indeed, the MBOA load was modified in AFA and AFB conditions, DIMBOA in AFB condition, and HDMBOA in AFA condition.

Forty-one hydroxycinnamic acids (HCA) and hydroxycinnamic amides (HCAA) were detected in wheat root extracts. Only seven compounds (17.1%) were significantly impacted by bacterial cells inoculation. Three derivatives had a significantly altered accumulation following inoculation with JV395B, two following inoculation with F113, and two following inoculation with both strains. Several sinapoyl, coumaroyl, and feruloyl derivatives could be detected, particularly sinapoyl-agmatine, feruloyl-putrescine, and coumaroyl-cadaverine. The content of sinapoyl-agmatine and feruloyl-putrescine was rather increased by the inoculation of both strains, while that of coumaroyl-cadaverine tended to decrease. Furthermore, inoculation of JV395B induces a differential production of sinapoyl-agmatine, feruloyl putrescine, coumaroyl cadaverine, and the coumaroyl derivative M273T479, depending on whether the bacterial cells were conditioned or not. This contrasting effect between conditioned and unconditioned cells on HCAA content is, however, not detected after F113 inoculation (Figure 6).

Another family of molecules whose production was affected by inoculation is the flavonoid family, of which, three compounds were detected in the extracts. These three flavonoids displayed characteristic UV spectra of apigenin, and their fragmentation suggested glycosylation on carbon atoms, corresponding to derivatives (apigenin-6-C-arabinoside-8-C-hexoside [24]), also called schaftoside and isoschaftoside [10]. The accumulation of two of the isomers was altered by inoculation. Inoculation of JV395B resulted in a significant decrease in schaftoside I production, especially when the cells were conditioned. There was also a strong tendency for a decrease in schaftoside II content, but this decrease was not significant, due to data variability. Schaftoside accumulation was also altered by inoculation of F113, particularly under AFA and AFC conditions (Figure 6).

The content of phenylalanine was also strongly reduced following inoculation of both strains, particularly that of the JV395B-conditioned cells (Figure 6).

## 3. Discussion

In this study, we evaluated the root metabolic responses of the two wheat genotypes Adular and Bordeaux to inoculation with two plant-beneficial strains, *P. chlororaphis* JV395B and *P. ogarae* F113 and we wondered whether the effects on wheat genotypes could be different if *Pseudomonas* cells have formerly been into contact with wheat genotypes.

Inoculation of each strain resulted in specific metabolic responses of each wheat genotype at the root level (Figure 2). Interestingly, each of the two genotypes responded differently to bacterial inoculations (Figure 2). In the Adular genotype, the two strains exhibited a contrasting effect and induced different changes in metabolite accumulation (Figure 2a). Whereas, for the Bordeaux genotype, the number of metabolites with modified production is smaller (Figure 2b). The specificities of the response of the genotypes to each of the strains may be influenced by several bacterial features. First, the phylogenetic distance between the strains may partly explain the differential observed responses. Indeed, the *P. ogarae* F113 strain belongs to the *P. corrugata* subgroup and JV395B to the *P. chlororaphis* subgroup; they may have different physiological behaviour and relationships with plants [4,19]. Second, the metabolites involved in plant-beneficial activity are different between the two strains [10,25]. Unlike JV395B, F113 is known to produce the antimicrobial polyketide compound, 2,4 diacetylphloroglucinol (DAPG) [10,25] and to have a significant plant-beneficial effect on several wheat genotypes [26], as well as on several other plants, such as maize or *Arabidopsis thaliana* [4,18]. Strain JV395B produces other bioactive secondary metabolites, such as phenazines or pyrrolnitrin [10,25]. Finally, strain JV395B is capable of producing at least six different AHLs, a feature that is not observed for strain F113 [10,25]. These data, thus, highlight a specificity of the strain × genotype interaction. Such a specificity has been demonstrated by inoculation of several phytostimulatory strains of the genus *Azospirillum* onto two maize genotypes [27] and on rice [28,29]. Similarly, a study on rice inoculated to a wider range of PGPR belonging to different genera showed different root metabolic profiles; however, this strain-specific response could not be fully related to the bacterial species, origin of inoculated strains, or their level of colonization, showing the complexity of these interactions [30].

A previous study showed that the production of secondary metabolites involved in the plant-beneficial properties of F113 and JV395B was strongly influenced by the root metabolites of Adular and Bordeaux genotypes [10]. Therefore, we sought to evaluate whether inoculation of F113 and JV395B cells conditioned by wheat root extracts resulted in a differential metabolic response in Adular and Bordeaux genotypes (Figure 1, Figure 3, Figure 4 and Figure 5). The Adular genotype was shown to respond more markedly to inoculation of conditioned cells than to inoculation of unconditioned cells (Figure 3 and Figure 4), particularly in the Adular × JV395B interaction (Figure 3). This differential interaction may be related to an increase in the accumulation of certain bacterial bioactive metabolites, when the JV395B cells are conditioned, notably the overproduction of phenazines or AHLs [10] (Appendix A). Indeed, AHLs are mediators that can have an inter-kingdom signaling effect. Besides their involvement in bacterial communication networks, they could be recognized by plant receptors and lead to the modification of plant gene expression [11]. The increase in AHL production in the conditioned cells can also be correlated with an increase in phenazine production [10,12]. Indeed, the conditioned JV395B cells produced a higher amount of phenazines, especially hydroxyphenazine (OH-PHZ) and dihydroxy-phenazine-1-carboxylic acid (di-OH-PCA) derivatives [10]. Phenazines are known to be major mediators of plant–bacteria interaction within the *P. chlororaphis* subgroup [15]. They play a major role in the survival of *P. chlororaphis* in the rhizosphere of wheat and in the protection of plant from pathogens, both via antagonistic activities, such as PCA on *Botrytis cinerea* and induction of induced systemic resistance (ISR) [15,16,17]. The pyrrolnitrin production is decreased in the conditioned JV395B bacterial cells. Pyrrolnitrin is an antimicrobial compound with a strong antifungal activity in vitro [25]. Bioactive secondary metabolites have both a direct positive effect on the plant itself [11,12,13,14,18] and indirect positive effect through their activity on other microorganisms in the rhizosphere [7,16,25].

The study of the Adular × F113 interaction also showed a gradient of responses, depending on whether the cells were conditioned or not (Figure 4). However, *P. ogarae* F113 conditioning did not result in an increase in mediators known to be involved in plant-beneficial activity. On the contrary, it led, for instance, to a decrease in the production of DAPG (Appendix A), which is a major compound responsible for the biocontrol activity of F113 [6]. Nevertheless, at sub-inhibitory concentrations, this molecule is known to have a signaling effect on plant roots by modulating certain biosynthetic pathways, such as the auxin hormonal pathway [18,19]. It is, therefore, conceivable that, at low concentrations, this molecule could induce metabolic changes in Adular roots. Moreover, conditioning of strain F113 cells significantly increased the production of compounds not yet studied for their potential plant-beneficial activity, such as pyridine-2,6-thiocarboxylic acid (PDTC) derivatives or lipophilic nitrogen derivatives (Appendix A, [10]). The changes of plant metabolic responses in the presence of the conditioned F113 cells highlights the interest of assessing the biological activity of these compounds that could be involved in strain × genotype interactions.

All these remarks must be tempered by the fact that the observed bacterial metabolic modifications were effective at the time of inoculation of the strains, but not necessarily after 21 days of culture, when collecting the plants and profiling their metabolic content. Nevertheless, the metabolic profiling of the two inoculated wheat genotypes demonstrated that a differential effect occurred, depending on whether the bacterial cells were conditioned or not (Figure 3, Figure 4 and Figure 5). It is also important to note that all of this experimentation was performed in non-sterile soil. Therefore, one can also suggest that the conditioned cells may have had a differential impact on the rhizosphere microbial community resulting in a specific metabolic response of wheat roots. Indeed, JV395B is able to produce AHLs, which are the main mediators of QS [11]. These mediators are ubiquitous and can mediate interspecies communication [31]. Many biological functions, such as motility, biofilm formation, or the production of bioactive compounds, such as antimicrobials, are under the control of QS in many bacterial genera [32,33]. We can, therefore, suggest that the extensive production of AHLs by JV395B may have altered the physiology and gene expression of several bacterial populations in the soil. Furthermore, although no AHLs were detected in the culture supernatant of strain F113 ([10,25] Appendix A), studies have shown that it could interact with other plant-beneficial bacteria, such as *Azospirillum baldaniorum* (previously *brasilense*) Sp245 [34], a biostimulant strain isolated from the wheat rhizosphere in Brazil [34,35]. Indeed, a low concentration of DAPG could trigger auxin production in Sp245 [35,36]. Inoculation of several *Azospirillum* strains on maize showed that these strains are also capable of inducing changes in maize root metabolism [37].

This response gradient was not observed for the Bordeaux genotype (Figure 5). However, the BJA condition (corresponding to the Bordeaux genotype inoculated with the Adular-conditioned JV395B cells) was separated from the other inoculated treatments on axis 2 of the PCA, suggesting a slightly different metabolic response of the plant, when the JV395B cells are conditioned with another extract than the Bordeaux genotype (Figure 5). This observation was also made in the Adular × F113 interaction, where the AFB condition is also separated along axis 2 of the PCA (Figure 4). This result is surprising because only a few differences were observed between the metabolisms of JV395B or F113 cells conditioned with Adular or Bordeaux extracts (Appendix A). One main difference between the JV395B cells conditioned by Bordeaux and the ones conditioned by Adular notably concerns the amount of phenazines [10]. One may wonder if, in the rhizosphere, there is not a potentiation of metabolic differences between bacterial cells conditioned by different plant extracts. One can also wonder if the wheat plant is able to differentially recognize cells conditioned by its own roots from cells conditioned by other roots, via signals not detected by the techniques implemented in this study.

All these data show complex metabolic modifications when the plant is inoculated with F113 and JV395B *Pseudomonas* strains, even more so when their cells are conditioned by plant root extracts. Many of the plant metabolites whose production is altered are minority compounds that are difficult to identify. Nevertheless, a molecular network approach allowed us to identify discriminant metabolites belonging to specific families (Figure 6). This approach was carried out on the Adular genotype, on which a significant separation between treatments after inoculation of F113 and JV395B strains was observed (Figure 3 and Figure 4). The metabolic network generated is very similar to the one presented in our previous work [10]. It contains mainly three large families of bioactive molecules, hydroxycinnamic acids (HCAs), hydroxycinnamic acid amides (HCAAs), benzoxazinoids, and flavonoids. Discriminant compounds could be highlighted in these three families of secondary metabolites, which represent the majority of active metabolites in wheat [38].

HCAs are derived from the metabolism of phenylpropanoids. The precursor of this pathway is phenylalanine, which is modified by phenylalanine amonia lyase (PAL), yielding cinnamic acid and, subsequently, other derivatives, such as coumaric, caffeic, or ferulic acid [39]. These derivatives then serve as precursors for the formation of other compounds, such as HCAAs or flavonoids. The decrease in phenylalanine accumulation in wheat roots after inoculation of the bacterial strains; specifically, JV395B conditioned cells may be explained by an increased processing through the phenylpropanoid metabolism pathway [39].

HCAAs are derived from a condensation between an HCA and an aliphatic polyamine, such as agmatine or putrescine [40]. They are common metabolites in the plant kingdom and are involved in several plant physiological processes, such as senescence or flowering [40]. These compounds are also involved in plant responses against biotic and abiotic stresses, such as chilling or pathogen attacks [40,41,42]. For example, the production of p-coumaroyldopamine and feruloyldopamine is increased after infection of tomato by the bacterial pathogen *Pseudomonas syringae* [43]. Another study, evaluating the metabolic response of wheat, also found that inoculation of the fungal pathogen *Bipolaris sorokiniana* resulted in an increase of 14 HCAAs in wheat leaves, including coumaroyl derivatives, such as coumaroyl-agmatine or coumaroyl-hydroxyputrescine [44]. In view of these results, the authors suggested an important role of HCAAs in the defense of wheat against *B. sorokiniana* and proposed a phytoalexin function in wheat for these compounds, similar to that of benzoxazinoids [44]. Another study was also able to show an increase in HCAAs accumulation in rice roots, but this time in the presence of plant-beneficial bacteria [30]. This study showed an increase in several HCAAs, such as N-p-coumaroyl putrescine or N-feruloylputrescine, in the presence of 10 PGPR, belonging to the *Azospirillum*, *Herbaspirillum*, and *Paraburkhoderia* genera, but a decrease in the presence of the pathogenic strain *Burkholderia glumae*, AU6208 [30]. The results obtained in our study are rather contrasting, with an increase in sinapoyl-agmatine and decrease in coumaroyl-cadaverine, triggered in Adular by strains F113 and JV395B, with differences depending on whether the cells are conditioned or not. All these results suggest that HCAAs probably play an important role in plant-microorganism interactions. Nevertheless, our study, associated with the literature data, shows the large number of HCA and HCAA derivatives that can be detected in wheat root extracts and the diversity of the activities of these molecules (Figure 6).

Another family of metabolites derived from HCAs, through a mixed biogenesis pathway, is the flavonoid family [39]. Flavonoids are also a highly represented family of secondary metabolites in the plant world [45], with more than 5000 different structures described so far [46]. Several have been reported in wheat, including schaftoside (apigenin-6-C-arabinoside-8-C-hexoside), which is one of the major compounds [24]. Flavonoids have been extensively studied in plant–bacteria interactions, especially in *Rhizobium*–legume mutualistic symbioses or actinorhizal symbiosis [45]. They also exhibit antimicrobial activities [47] and activity on rhizospheric bacteria [48,49,50,51]. Two isomers of schaftoside were detected in root extracts, likely schaftoside and isoschaftoside (Figure 6). Both of these derivatives tend to be decreased following inoculation with F113 and JV395B, and this decrease is particularly significant for schaftoside I following the inoculation of conditioned JV395B cells (Figure 6). The activity of flavonoids on bacterial strains can be specific, depending on the strain and the compound [48]. To our knowledge, no study has specifically evaluated the impact of schaftoside on *Pseudomonas* strains. Moreover, some flavonoids, such as naringenin, can interfere with QS communication in some bacterial strains [50]. We can hypothesize that the conditioned JV395B cells, which produce higher levels of AHLs in the presence of wheat root extracts (Appendix A), may interfere with flavonoid production in the Adular genotype. It is also noteworthy that the HCA and flavonoid biosynthetic pathways are interconnected [51]. For example, flavonoid production is altered in maize genotypes deficient in the benzoxazinoid pathway [52]. It can, therefore, be hypothesized that the decrease in schaftoside may not be due to the direct impact of *Pseudomonas* inoculation but could result from the altered production of other wheat metabolites.

Benzoxazinoids represent the family of molecules containing the most discriminating compounds. These plant secondary metabolites are found in the *Poaceae* and some other plant families [53]. They are particularly studied in wheat, maize, and barley, in which they are produced during the early stages of plant growth [54,55]. They represent one of the major classes of phytoalexins in the *Poaceae* [55,56], acting particularly by protecting young plants from insect and pathogen attacks [57,58]. Interestingly, benzoxazinoids trigger various effects on microorganisms [52,54,57,58,59]. For example, DIMBOA can exert a toxic effect on a pathogenic strain of the genus *Agrobacterium*, as well as a positive attractive effect on a *Pseudomonas putida* strain, accompanied by a potentiation of its phytoprotective activity [54,60]. Comparison of the rhizomicrobiota from a wild-type maize with that of a BX minus genotype, unable to produce benzoxazinoids, also showed that the two genotypes did not recruit the same community [52,57]. Interestingly, the microbiota selected by the genotype capable of producing benzoxazinoids exerts a protective effect on the nextgeneration plant, while this is not the case for the rhizomicrobiota selected by BX minus genotype [57]. This work shows a role for benzoxazinoids in the interaction of maize with its rhizomicrobiota, as well as a role in selecting the microorganisms capable of protecting plants [57]. The benzoxazinoid biosynthetic pathway starts from tryptophan and results in several glycosylated and non-glycosylated derivatives [53]. In our study, inoculation of F113 and JV395B strains on the Adular genotype tended to result in a decrease in glycosylated derivatives (i.e., DHBOA-Glc, DIBOA-Glc, HMBOA-Glc, and DIMBOA-Glc) that is significant after inoculation of the conditioned JV395B cells. Glycosylated derivatives are considered as storage forms, contained in the vacuole of plant cells [61]. Upon tissue degradation, they are released into the external environment and are clived by β-glucosidases releasing the more active aglycone forms [61]. Inoculation of conditioned JV395B cells induced an increase in putative HDMBOA content and strong tendency to increase the MBOA content (univariate test, *p*-value = 0.12). In the case of the F113 strain, the amount of putative HDMBOA is increased only in the AFA condition and that of MBOA elevated following inoculation of cells conditioned by both Adular and Bordeaux (compared to the uninoculated condition). It can be noted that HDMBOA is the aglycone form that is most rapidly transformed into MBOA [61]. Aglycone derivatives and MBOA are referenced in some studies as the most active forms of benzoxazinoids (Hu et al., 2018). MBOA alone, for example, can conditionate the maize microbiota and, in turn, lead to a protective effect on the next generation of plants [57]. This change in the balance between glycosylated and aglycone forms, as well as the increase in MBOA following inoculation of the *Pseudomonas* strains (especially the conditioned cells, compared to the uninoculated condition) may, therefore, have a significant effect on the physiology of the genotypes and their biotic interactions with the rhizomicrobiota.

Altogether, the results of this study showed that bacterial strains induced specific metabolic responses in each of the two wheat genotypes and supports previous studies, which suggest that the interaction was specific. They also showed that the wheat genotypes responded differently to inoculation with the conditioned and unconditioned cells of *P. ogarae* F113 and *P. chlororaphis* JV395. These results suggest that the plant differently recognizes the bacterial cells that have already been into contact with plants. This work was performed on two wheat genotypes and two *Pseudomonas* strains. Performing this experience on several other biological models could allow to more deeply understand specific interaction between plant host and PGPR. Moreover, this work needs to be further investigated, in order to identify the elements involved in this differential recognition of conditioned bacterial cells. Nevertheless, our study demonstrates the ecological feedback between wheat and *Pseudomonas* rhizospheric bacteria and highlights new avenues to be considered in the preparation of bacterial inocula in agriculture.

## 4. Materials and Methods

### 4.1. Biological Material

This study was performed on two PGPR, belonging to the fluorescent *Pseudomonas* group, *P. ogarae* F113 isolated from a sugarbeet rhizosphere [62], and *P. chlororaphis* JV395B isolated from the soil of fields cultivated with corn [4]. The bread wheat (*Triticum aestivum* L.) genotypes are Bordeaux 113 (7973) and Adular (797) (source: UMR GEDC, INRA Clermont-Ferrand).

### 4.2. Conditioning of Bacterial Strains before Inoculation

The bacterial strains *P. ogarae* F113 and *P. chlororaphis* JV395B were previously conditioned with the root extracts of the Adular and Bordeaux genotypes, obtained according to the same protocol as the previous study [10] (Figure 1a). Conditioning was performed in MM fructose medium, supplemented with root extracts of the Adular and Bordeaux genotypes at a concentration of 50 µg·mL^−1^ (1% methanol/ 99% water), according to the protocol presented in previous work [10]. Methanol was used as a control condition. Each of the six conditions were performed in 30 different wells of a Bioscreen plate (*N* = 30), and the growth of the strains was measured over a period of 54 h.

After 54 h of culture (i.e., incubation time allowing production and detection of a significant diversity of *Pseudomonas* metabolites), 200 µL of six different wells were randomly transferred and pooled in Eppendorf^®^ (Hamburg, Germany) to form five replicates per condition (*N* = 5). The 1200 µL (6 × 200 µL) of culture were then centrifuged (10 min, 5500 rpm), in order to separate the culture medium from the bacterial cells. The supernatant was transferred to a new Eppendorf^®^, in order to extract the bacterial secondary metabolites via a liquid/liquid extraction, according to a previously described protocol [10], and then analyzed by LC–HRMS. The bacterial pellets were recovered in 10 mM magnesium sulfate (MgSO4) and pooled to form a final inoculum at 2 × 10^7^ CFU·mL^−1^.

### 4.3. Bacterial Inoculation of Plants and Plant Growth

Conditioned and unconditioned bacteria were then inoculated onto the two wheat genotypes (Figure 1b). Seeds of the Adular and Bordeaux genotypes were washed by shaking in 40 mL of distilled water (8 min; 50 rpm). They were then transferred to 2 dm^3^ pots, containing unsterilized, sieved soil (Ø 4 mm) from a luvisol originating from an experimental farm (La Côte Saint André, Isère, France). Fifty microliters of a bacterial inoculum (i.e., 10^6^ CFU) were then deposited on the seeds for each condition (Figure 1b). Fifty microliters of 10 mM MgSO4 were used for the non-inoculated condition.

Each of the pots contained three seeds, and each of the conditions were performed in eight different pots (i.e., 24 plants/condition). Then, the pots were arranged randomly, with a rotation every two days, under 16 h day and 8 h night photoperiods, at a temperature of 20 °C. All along the culture, the soil was moistened at a water content of 20% (*v*/*v*) and the growth of the aerial part (i.e., length of aerial parts between the crown and the longest leaf) was monitored.

Twenty-one days after inoculation, the wheat plants were harvested. The plants were slightly shaken to remove the soil not adhering to the roots; then, the aerial parts were separated from the root parts. In a pot containing three plants, one plant was used for root and leaf dry biomass measurements after 24 h of drying in an oven at 100 °C. The last two plants were used for the analysis of root secondary metabolites. For this purpose, they were directly placed in liquid nitrogen to block any enzymatic reactions and lyophilized for 72 h at −54 °C (Alpha 1–4 LSC Christ, Osterode, Germany). The dry roots were then ground in the presence of metal balls after immersion in liquid nitrogen using a vibro-grinder (TissueLyser II; Qiagen, Courtabœuf, France). Before extraction, two roots were randomly pooled to form a total of seven replicates per condition (*N* = 7). Extraction of root metabolites was then performed according to the methanol extraction protocol used in previous work [10]. Each extract was then resolubilized in methanol to a final concentration of 10 mg·mL^−1^ before being analyzed by LC–HRMS.

### 4.4. High Performance Liquid Chromatography Analysis Coupled with High Resolution Mass Spectrometry

Plant and bacterial extracts were analyzed on an Agilent technologies^®^ (Santa Clara, CA, USA) Accurate-Mass Q-TOF LCMS 6530 instrument, coupled with an LC 1290 Infinity system, according to the analytical methods, adapted to bacterial and plant extracts, as previously described [10].

### 4.5. Data Processing and Statistical Analysis

The metabolomic analysis was focused on small molecules, with weights below 1000 Da, for both bacterial and plant extracts. The workflow used for pre-processing was similar to that used previously [10], but with a slight modification of the parameters. The parameters used in this study are described in Appendix A. Statistical analysis of the impact of conditioning by plant extracts on the bacterial metabolism was performed using the same protocol described in previous work [10].

A first statistical analysis was performed to evaluate the effect of unconditioned bacterial cells inoculation on wheat root metabolome. Then, a second statistical analysis allowed us to evaluate specifically conditioned bacterial cells effects on wheat root metabolome. Statistical analysis of metabolic changes in wheat, following strains inoculation, were performed in several steps to evaluate both inoculation and conditioning effect. First, a separation of the conditions was performed by a discriminant analysis (“PLS discriminant analysis”, PLS-DA) via the rolps package [63]; this allowed the identification of metabolites involved in the separation of the conditions. A univariate analysis was then performed on these metabolites by a non-parametric Kruskal-Wallis test, followed by a post hoc test with an FDR correction via the agricolae package [64]. Finally, a principal component analysis (PCA) was performed on the discriminant compounds, in order to visualize the differences between conditions via the ade4 package [65].

When analyzing the wheat metabolic changes induced by the conditioned and unconditioned strains, a clustering of the conditions (i.e., “clustering”) was performed according to the Euclidean distance via the permutMatrix tool [66]. Metabolites sharing the same behaviour were manually grouped, according to the results of univariate statistical tests, showing their differences between conditions.

### 4.6. Molecular Network Analysis and Metabolite Identification

The molecular network was performed via the Metgem software [67], after data pre-processing via Mzmine 2 v2.53 software, following the same protocol as previously described [10], using parameters described in Appendix A. The molecular networking approach allowed us to cluster the compounds belonging to the same chemical family. Thus, the identification of metabolites can be achieved by comparison to commercial standards, allowing for level 1 identification, according to Sumner et al., confidence annotation level (Appendix A). The identification was also carried out by comparing metabolites UV spectra, accurate masses, and MS/MS fragmentations to those of standards and/or to literature data, which allowed for level 2 and 3 identifications. More detailed information is available in our previous work on wheat [10] and *Pseudomonas* [25] metabolites.

## Figures and Tables

**Figure 1 metabolites-12-00236-f001:**
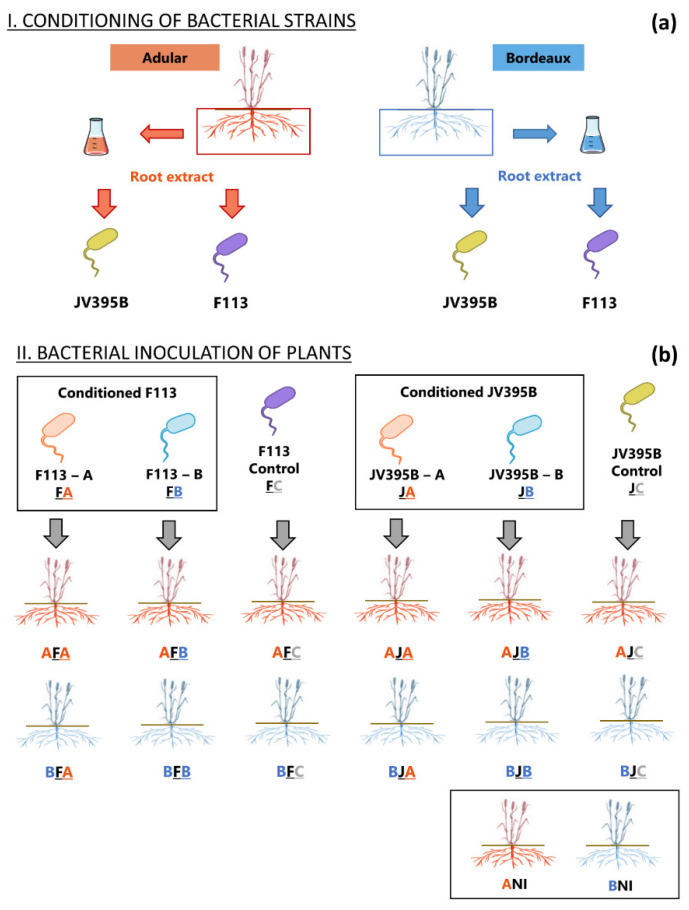
Experimental setup for conditioning of *Pseudomonas* strains (**a**) and inoculation of wheat genotypes (**b**). Root extracts from Adular and Bordeaux wheat genotypes (50 µg·mL^−1^) were put in contact with the bacterial cells of *P. ogarae* F113 and *P. chlororaphis* JV395B in a liquid minimal medium (**a**). Then, after a 54h-incubation, conditioned and unconditioned *Pseudomonas* cells were inoculated on Adular and Bordeaux genotypes, leading to 14 different conditions (**b**).

**Figure 2 metabolites-12-00236-f002:**
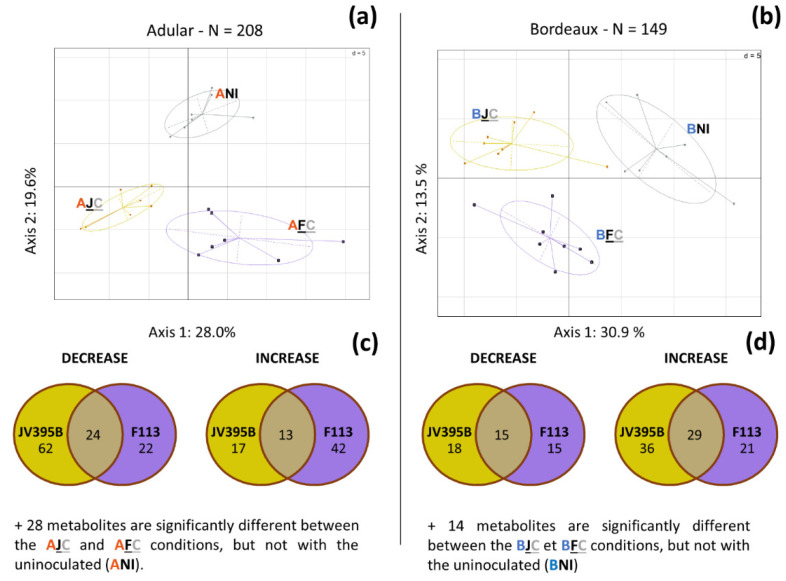
Impact of inoculation of the unconditioned F113 and JV395B cells on root secondary metabolism of Adular (**a**,**c**) and Bordeaux (**b**,**d**) wheat genotypes. Principal component analysis (PCA) (**a**,**b**) and Venn diagrams (**c**,**d**) were performed using only the 208 and 149 discriminant metabolites from Adular and Bordeaux, respectively. The PCA displays the separation between the different conditions (**a**,**b**) (ANI: uninoculated Adular; AFC: Adular inoculated with unconditioned F113; AJC: Adular inoculated with unconditioned JV395B; BNI: uninoculated Bordeaux; BFC: Bordeaux inoculated with unconditioned F113; BJC: Bordeaux inoculated with unconditioned JV395B). Venn diagrams represent the distribution of metabolites according to the different conditions (**c**,**d**).

**Figure 3 metabolites-12-00236-f003:**
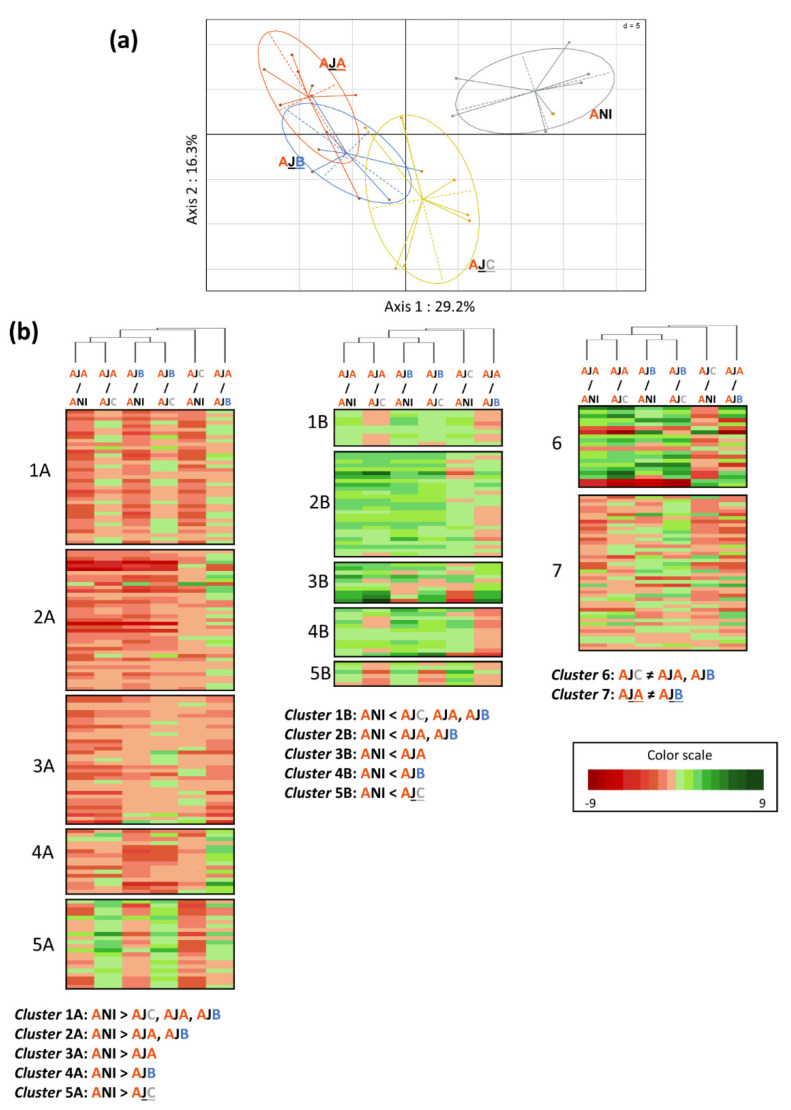
Impact of the conditioned and unconditioned *P. chrororaphis* JV395B cells on Adular root secondary metabolism. PCA (**a**) and heatmap (**b**) were made only on the 277 discriminant metabolites, obtained after LC-HRMS analysis of the Adular root extracts. The PCA represents the separation between the different conditions (**a**) (ANI: uninoculated Adular; AJA: Adular inoculated with Adular-conditioned JV395B; AJB: Adular inoculated with Bordeaux-conditioned JV395B; AJC: Adular inoculated with unconditioned JV395B). The heatmap shows the clustering of discriminant Adular metabolites into 12 different groups, according to their specific behaviour between conditions (**a**). Cluster 1 represents metabolites whose accumulation significantly decreased A or increased B after JV395B inoculation, whatever the type of cells conditioning. Cluster 2 represents metabolites that significantly decreased A or increased B after inoculation of conditioned JV395B cells only. Cluster 3 represents metabolites significantly decreased A or increased B after inoculation of Adular-conditioned JV395B cells. Cluster 4 represents metabolites significantly decreased A or increased B after inoculation of Bordeaux-conditioned JV395Bcells. Cluster 5 represents metabolites significantly decreased A or increased B after inoculation of unconditioned JV395B cells. Cluster 6 represents metabolites with opposite behaviour according to inoculation of conditioned or unconditioned JV395B cells and then, cluster 7 represents metabolites with different behaviour, depending on whether JV395B was conditioned by a root extract of Adular (AJA) or Bordeaux (AJB).

**Figure 4 metabolites-12-00236-f004:**
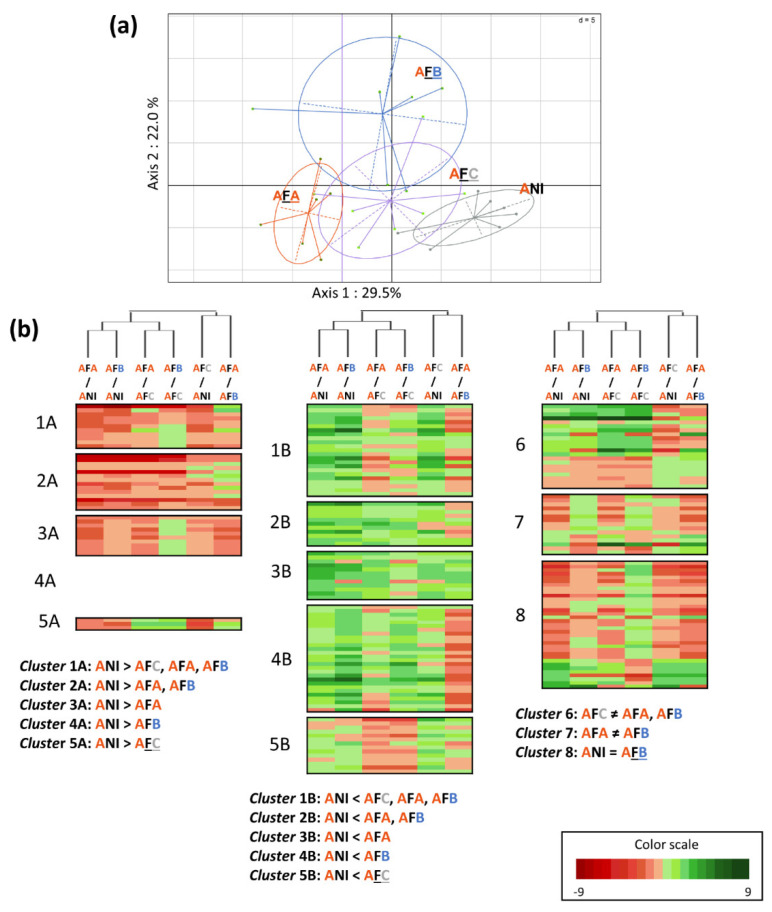
Impact of the conditioned and unconditioned *P. ogarae F113* cells on Adular root secondary metabolism. PCA (**a**) and heatmap (**b**) were made only on the 197 discriminant metabolites obtained after LC-HRMS analysis of Adular root extracts. The PCA represents the separation between the different conditions (**a**) (ANI: uninoculated Adular; AFA: Adular inoculated with Adular-conditioned F113; AFB: Adular inoculated with Bordeaux-conditioned F113; AFC: Adular inoculated with unconditioned F113). The heatmap shows the clustering of Adular discriminant metabolites into 12 different groups, according to their specific behaviour between conditions (**b**). Cluster 1 represents metabolites whose accumulation is significantly decreased A or increased B after F113 inoculation, whatever the type of cells conditioning. Cluster 2 represents metabolites significantly decreased A or increased B after inoculation of conditioned F113 cells only. Cluster 3 represents metabolites significantly decreased A or increased B after inoculation of Adular-conditioned F113. Cluster 4 represents metabolites significantly decreased A or increased B after inoculation of Bordeaux-conditioned F113. Cluster 5 represents metabolites significantly decreased A or increased B after inoculation of unconditioned F113 cells. Cluster 6 represents metabolites with opposite behaviour, according to inoculation of conditioned or unconditioned F113 cells. Cluster 7 represents metabolites with different behaviour, depending on whether F113 cells were conditioned by a root extract of Adular or Bordeaux and then, cluster 8 represents metabolites, with similar behaviour between the uninoculated condition (ANI) and Bordeaux-conditioned F113 cells (AFB condition).

**Figure 5 metabolites-12-00236-f005:**
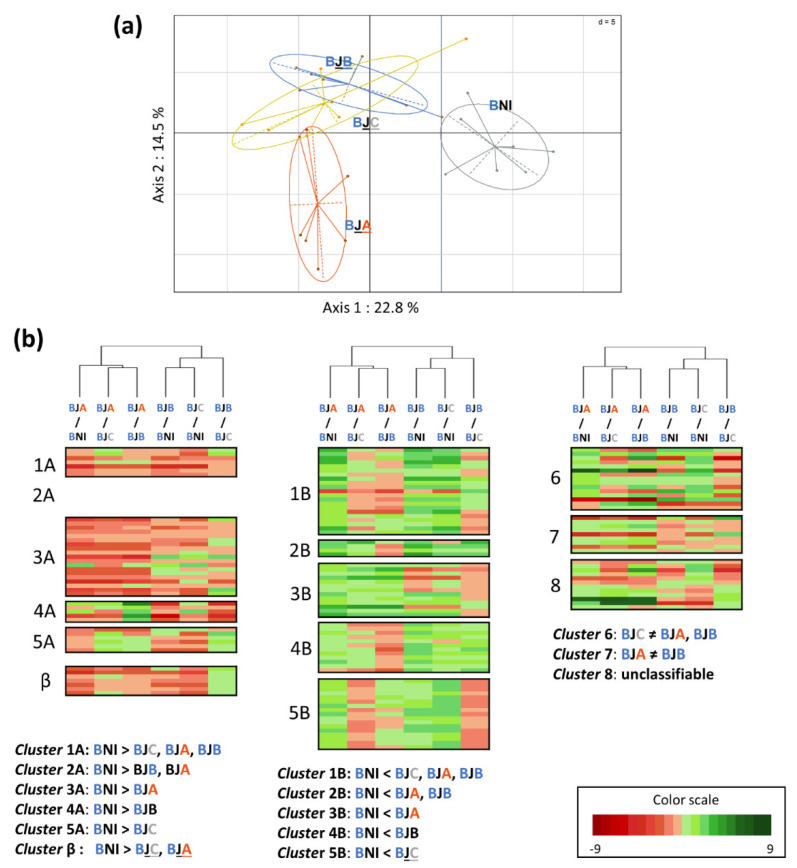
Impact of the conditioned and unconditioned *P. chrororaphis* JV395B cells on Bordeaux root secondary metabolism. PCA (**a**) and heatmap (**b**) were made only on the 147 discriminant metabolites obtained after LC-HRMS analysis of Bordeaux root extracts. The PCA represents the separation between the different conditions (**a**) (BNI: uninoculated Bordeaux; BJA: Bordeaux inoculated with Adular-conditioned JV395B; BJB: Bordeaux inoculated with Bordeaux-conditioned JV395B; BJC: Bordeaux inoculated with unconditioned JV395B). The heatmap shows the clustering of Bordeaux discriminant metabolites into 12 different groups, according to their specific behaviour between conditions (**b**). Cluster 1 represents metabolites whose accumulation is significantly decreased A or increased B after JV395B inoculation, whatever the type of cells conditioning. Cluster 2 represents metabolites significantly decreased A or increased B after inoculation of conditioned JV395B cells only. Cluster 3 represents metabolites significantly decreased A or increased B after inoculation of Adular-conditioned JV395B cells. Cluster 4 represents metabolites significantly decreased A or increased B after inoculation of Bordeaux-conditioned JV395B cells. Cluster 5 represents metabolites significantly decreased A or increased B after inoculation of unconditioned JV395B cells. Cluster 6 represents metabolites with opposite behaviour, according to inoculation of conditioned or unconditioned JV395B cells; then, cluster 7 represents metabolites with different behaviour, depending on whether JV395B was conditioned by root extract of Adular (BJA) or Bordeaux (BJB); then, cluster 8 represents metabolites with atypical behaviour that cannot be categorized into other clusters.

**Figure 6 metabolites-12-00236-f006:**
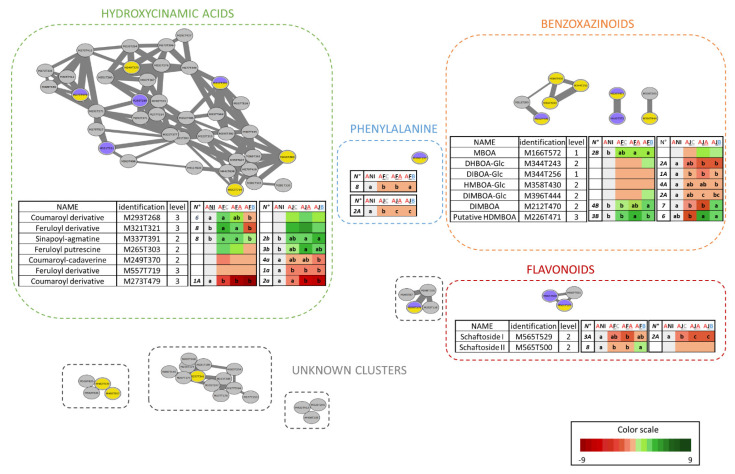
Molecular networks of MS/MS data obtained from inoculated and uninoculated Adular root extracts. Molecular network is associated with heatmaps representing discriminant metabolite accumulation, according to the conditions and their cluster number on untargeted analysis. The “level” column represents the confidence level of the metabolite annotation. Node colours represent the discriminant metabolites after *P. ogarae* F113 (purple) and *P. chlororaphis* JV395B (yellow) inoculation. Statistical discrimination (Kruskal-Wallis test; *p*-value < 0.05) is indicated by the letters in the heatmaps.

## Data Availability

The data presented in this study are available on request from the corresponding author. The data are not publicly available due to privacy.

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
