# Peer review of "Wheat Metabolite Interferences on Fluorescent Pseudomonas Physiology Modify Wheat Metabolome through an Ecological Feedback"

_metabolites, 2022, doi:10.3390/metabo12030236_

Round 1
Reviewer 1 Report
In the reviewed manuscript, the Authors present results of studies on the influence of Pseudomonas strains, conditioned by root metabolites of two wheat genotypes, on the root metabolome of these wheat genotypes. The reviewed manuscript may be considered as a continuation of the previous work published by the Authors in 2021 (Metabolites 2021, 11, 84), where the impact of the wheat root metabolites on the Pseudomonas strains metabolome was analyzed.
In the current study the Authors put the hypothesis that 'Pseudomonas cells that have been previously in contact with wheat root extracts will produce different secondary metabolites in the rhizosphere and, consequently, will then differentially impact the plant physiology compared to cells that have not been previously in contact with the plant.' To test this hypothesis the Authors used very well documented approach and methodology. The results are described in details and supported with numerous figures, diagrams and tables. The authors clearly demonstrate that bacterial strains induce specific metabolic responses in each of the two wheat genotypes.
In my opinion, the discussion is comprehensive and maybe even overloaded to some extent with detailed information on the identified microbial metabolite groups and their putative impact on the plant physiology/growth, which of course was not the aim of the study and was not tested during the research, but it can guide future studies.
Minor comments:
I would be grateful to the Authors for answering my question regarding the information provided in lines 602-603 - why did the Authors choose 54h? There is no indication as to why this particular period was selected.
line 610 - there should be 2 x 10(7 in superscript) x ml(-1 in superscript) or 2 x 10 (7 in superscript) / ml
Reviewer 2 Report
The manuscript "Wheat Metabolite Interferences on Fluorescent Pseudomonas Physiology Modifies Wheat Metabolome through an Ecological Feedback" reports a very interesting and new study about the effect of two beneficial strains belonging to Pseudomonas genus on two wheat genotypes in terms of changes in metabolic profiles. It is a certainly new and less explored field as approached with metabolomics methodologies.
Introduction provide a sufficient background and it makes clear the novelty of the study. It could be make more complete by adding more informations of beneficial effects exerted by the selected strains (F113 and JV395B) on plants, as it was done in discussion section.
Methods are the same used for a previous experiment and so it's reported the reference in which it's possible to find all the informations necessary to repeat the experiment. Minor adjustement are reported in the attached pdf file.
Results are adequately described and reported in most sections.
Figures in the manuscript contains information that are missing in the text (i.e. the number of discriminant metabolites, or the color of nodes in molecular networks); also the description of clusters is less clear compared to the text. The same set of informations should be written both in the text and in figure captions, so that the reader is able to read figures independently from the text and viceversa.
Discussion supports results obtained with a good number of references. But it's easy to get lost while reading because the first part deals with results not displayed in the text but in supplementary material, that are also very similar to the ones obtained in the previous work published by the authors. Results section mostly deals with the effect of the strains (both conditioned or not) on wheat metabolome, but the discussion often refers to the effect of the conditioning on bacterial cells, which makes the section a bit of confusing. It is clear that there is a correlation between the response of bacterial cells to conditioning and the response of wheat roots to the strains differently conditioned; but throughthout the text the conditioning of F113 and JV295B is presented as a minor part of the work (since previous results obtained from the same authors), so it should have less space in discussion sections and also in conclusions (lines 582-587). Conclusions should deal more with the aim of the work stated at the end on the introduction, that is specifically focused on wheat metabolome and not strains metabolic changes after conditioning.
Other comments are reported in the attached pdf file.

Reviewer 3 Report
Dear Authors, I have read your manuscript with a good deal of zeal and zest. I found your manuscript interesting and worth publishing just after minor corrections.
The scientific problem has been well addressed in introduction however, there is a need to have a gap statement.
How this work differes among the mentioned ones.
The materials and methods section seems okay, however i suggest elaborating a bit on biological assays for metabolite determinations.
since, the strains were able to produce AHLs, i wonder if they can pose damage to wheat crop? Have the authors had host specificty test? Werte these strains host specific?
I am concerned with their activity in rhizosphere, were they able to shift their activity in non-host rhizosphere?
The limitations should be mentioned in conclusion and also provide a separate conlusion section.
